# Organ pedalboard as a rehabilitation tool: A qualitative exploratory study of healthcare providers' perceptions and recommendations

**Mandy M. P. Kan** [1], **Wai Hang Kwok** [2], **Eric C. Fan** [3], **Fadi M. Al Zoubi** [1] *

**1** Department of Rehabilitation Sciences, Faculty of Health and Social Sciences, The Hong Kong Polytechnic University, Hong Kong SAR, China, **2** School of Nursing and Midwifery, Edith Cowan University, Western Australia, Australia, **3** Hong Kong Institute of Music Plus Limited, Hong Kong SAR, China

* fadi.alzoubi@polyu.edu.hk

## Abstract

### Objective

This qualitative study explored healthcare providers' perceptions and attitudes regarding the use of organ pedalboards as a rehabilitation tool, particularly for lower extremity conditions. The study also sought to identify the perceived barriers and facilitators to adopting organ pedalboards within rehabilitation settings, as well as gather healthcare providers' recommendations for integrating this tool into clinical practice.

### Method

Healthcare providers, including physiotherapists, occupational therapists, music therapists, and sports therapists, were recruited for focus group interviews using purposive and snowball sampling. At the outset of the interview, participants watched an organist playing musical notes on a pedalboard and had the opportunity to try on the pedalboard. A researcher then modulated the group discussion with the organist and asked questions following a semi-structured interview guide. The guided questions concern the perceptions of using organ pedal training for patients, as well as participants' attitudes and perceived barriers towards it. Verbatim transcription and content analysis were performed on qualitative data.

### Findings

Seventeen healthcare providers were interviewed in four focus groups. Healthcare providers perceive music as a motivator in rehabilitation, aiding in natural movement through rhythm and beats. Music intervention, particularly training on the pedalboard, is seen as beneficial for lower extremity rehabilitation, improving range of motion, balance, and coordination. It also offers cognitive benefits and enhances patient engagement and psychosocial well-being. Its safety concerns were addressed and discussed.

**Data Availability Statement:** All relevant data are within the manuscript and its Supporting information files.

**Funding:** The author(s) received no specific funding for this work.

**Competing interests:** The authors have declared that no competing interests exist.

## Conclusion

Our study is the first to explore the feasibility of using a pedalboard as a rehabilitation tool. Healthcare providers identify the circumstances and potential therapeutic benefits of the use of organ pedal training in the management of lower extremity problems. This will assist in the development of a lower extremity training protocol that can accommodate all the physiological hip, knee, ankle, and foot movements in the future.

## Introduction

The therapeutic impact of music on health and well-being has gained significant attention in contemporary research. Music has been shown to divert individuals from unpleasant experiences and facilitate physical movement, social interaction, and communication [1]. Positive music experiences have been associated with increased emotional well-being and dopamine activity in the brain, leading to reduced anxiety, worries, and agitation [2]. Additionally, music's influence on the sympathetic systems of the autonomic nervous system has physiological effects, including modulation of heart rate variability, reduction in heart rate and blood pressure, and enhancement of immune system function [3–6].

In rehabilitation, music intervention has become a vital component, extending beyond traditional therapy. It involves the strategic use of musical elements, integrated with standard exercises and treatments [2, 7, 8]. The therapeutic effects of music-based interventions have been extensively studied, particularly in relation to neurorehabilitation, where such interventions have demonstrated efficacy in improving motor and cognitive functions [9, 10]. Our focus on lower extremity conditions arises from the observation that rehabilitation of these conditions, such as stroke-related gait impairments, often necessitates interventions that are both motivating and conducive to sustained engagement. While lower extremity conditions encompass a wide range of impairments, including those affecting balance, gait, and overall mobility [11], this diversity underscores the need for innovative and versatile rehabilitation tools capable of addressing various challenges within this category. Music-based interventions, particularly those that incorporate rhythm and movement coordination, have shown promise in enhancing rehabilitation outcomes for these impairments, making them a compelling area for further research [12–14].

Several studies supported the role of music intervention in lower limb rehabilitation. For instance, a randomized controlled trial demonstrated that music intervention coupled with rehabilitation exercises significantly alleviated pain and improved mobility and gait in children with lower limb burns [15]. Another study [16] investigated the effects of self-selected versus motivational music on muscle endurance and affective state in middle-aged adults. The study employed a within-subject design where 26 healthy middle-aged males (average age 50.8±8.4 years) performed maximal and endurance isometric strength tests under three different conditions: self-selected music (SSM), motivational music (MM), and a control condition (CC) with no music. The results showed that mean force during the isometric endurance test was significantly higher in the SSM condition (507.3±132.2 N) compared to both MM (476.3±122.4 N, p<0.01) and CC (484.6±119.2 N, p = 0.03). Additionally, participants reported a higher Felt Arousal Scale (FAS) score and Feeling Scale (FS) in the SSM condition, indicating a more positive affective state. These findings suggest that self-selected music can enhance muscle endurance and positively influence mood during strength exercises in middle-aged adults. These findings underscore the potential of music intervention to divert attention from discomfort

and reduce perceived exertion levels. For instance, Magill-Levreault [17] highlighted the role of music therapy in pain management, where it was shown to decrease pain perception by altering affective and cognitive processes. Similarly, Mohammadzadeh et al. [18] found that music significantly reduced the rate of perceived exertion (RPE) during progressive exercise, especially among untrained individuals, while Potteiger et al. [19] demonstrated that various types of music, including self-selected music, were associated with lower RPE during moderate-intensity exercise. These studies collectively highlight the prospective utility of music in rehabilitation settings, where managing discomfort and perceived exertion is crucial.

Amid the growing use of musical instruments in rehabilitation interventions, this study explored a novel area: the use of organ pedal training for lower extremity rehabilitation. Evidence supporting the use of organ pedaling as an effective intervention for lower extremity rehabilitation comes from various domains where music and movement are integrated into therapeutic practices. Organ pedal training, which involves coordinated lower limb movements similar to those required in other rhythmic and music-based therapies, has shown promise in enhancing motor functions and overall rehabilitation outcomes. For instance, music therapy has been widely recognized for its positive impact on motor recovery in neurological conditions such as stroke and Parkinson's disease, where rhythmic auditory stimulation significantly improves gait and mobility [20, 21]. This suggests that the rhythmic and repetitive nature of organ pedaling could similarly benefit lower extremity rehabilitation. Moreover, the physical demands placed on organists, who frequently engage in complex foot movements, are comparable to those seen in athletic training. This has led to observations that organists develop enhanced lower limb strength, endurance, and coordination, further supporting the potential of organ pedalboard as a rehabilitation tool [22]. Additionally, music's ability to serve as a motivator during physical activity, reducing perceived exertion and increasing endurance, has been well-documented in exercise settings [18, 19]. These findings are indicative of the broader applicability of music-integrated physical exercises in promoting lower limb functions. Although direct studies on organ pedaling are limited, the extrapolation from related music and movement therapies strongly supports its potential effectiveness in treating lower extremity conditions. The unique combination of rhythmic movement and cognitive engagement in organ pedaling makes it a promising area for further research and application in rehabilitation settings.

The selection of the organ pedalboard as a rehabilitation tool is grounded in its unique ability to engage the lower extremities in a musically and rhythmically structured environment. Unlike many other musical instruments, the organ pedalboard requires precise, coordinated foot movements to produce sound, thereby directly targeting motor control and coordination in the lower limbs [23]. This makes it particularly well-suited for rehabilitation applications. The repetitive, goal-oriented nature can improve functional outcomes through tasks that mimic daily activities. Despite limited research in this specific domain and the recognized need for more musical interventions for lower extremity rehabilitation, organ pedal training holds potential as a non-invasive and possibly beneficial approach.

To explore the therapeutic potential of organ pedaling for lower extremity rehabilitation, this study retrospectively applied the Biopsychosocial Model [24] as a guiding conceptual framework. The Biopsychosocial Model proposed by Engel (1977) posits that health outcomes result from the intricate interplay between biological, psychological, and social factors. This framework is particularly apt for understanding how organ pedaling might influence lower extremity conditions, as it addresses not only the physical rehabilitation of muscles and joints but also the psychological and social dimensions of recovery. From a biological perspective, organ pedaling engages multiple lower limb muscle groups, promoting range of motion, flexibility, and strength, which are critical for rehabilitation [20]. Psychologically, the rhythmic and

musical nature of pedaling can enhance patient motivation and adherence to rehabilitation protocols by reducing the perception of effort and increasing overall engagement [19]. Socially, the use of music-based interventions such as organ pedaling may facilitate social interaction, improving the overall well-being of patients through shared musical experiences [25]. By incorporating the biopsychosocial model, this study aims to provide a comprehensive understanding of the potential benefits of organ pedaling for lower extremity rehabilitation. This framework guides the research process, ensuring that the investigation remains focused on the interconnected biological, psychological, and social factors that contribute to successful rehabilitation outcomes.

## Method

### Design

This qualitative study employed focus group interviews to explore healthcare providers' perceptions of using organ pedalboards as a rehabilitation tool for lower extremity conditions. The decision to use focus group interviews was intentional, as this method allows participants to interact, build on each other 's ideas, and generate deeper insights through group dynamics. Furthermore, the focus groups were designed to include healthcare providers from different disciplines, such as physiotherapists, occupational therapists, music therapists, and sports trainers. This interdisciplinary composition was a deliberate choice, as it was expected to foster diverse perspectives and encourage cross-disciplinary dialogue, yielding a richer understanding of the potential benefits and challenges of integrating organ pedalboard into rehabilitation practice. By bringing together professionals with varied expertise and backgrounds, the interdisciplinary focus group discussions were anticipated to uncover nuanced insights that may not have emerged from individual interviews or a more homogeneous group, stimulating new ideas, challenging assumptions, and providing a comprehensive exploration of the topic from multiple perspectives.

This study was conducted and reported according to the consolidated criteria for reporting qualitative research (COREQ) checklist for qualitative research (S1 File) [26].

### Participants and sampling

Purposive and snowball sampling were employed to recruit healthcare providers from four disciplines—physiotherapists, occupational therapists, music therapists, and sports trainers—until data saturation reached [27]. To address the broad concept of "lower extremity conditions" and achieve data saturation, we continued sampling until no new themes emerged from the interviews. This was monitored closely through ongoing data analysis and was confirmed when subsequent interviews added no new significant insights.

Participants had to be currently practicing full- or part-time at hospitals, non-governmental organizations, or in private practice. No prior experience with organ pedalboards was required for participation, as the primary focus was on exploring perceptions and potential applications of organ pedal training rather than assessing technical proficiency. Those unable to communicate in either Chinese or English were excluded. Potential participants were approached in September and October 2023 through telephone or email by the lead researcher, MK. Few participants knew MK personally, a female research associate (registered nurse; MA in psychology in music) prior to the research. The study aims and procedure and the experience and interest of the research team were introduced to all participants. All participants gave written informed consent prior to inclusion. The consent process detailed the study's aims, procedures, potential risks, and benefits. Participants were also advised of their right to withdraw at any time without repercussions.

The rationale for selecting physiotherapists, occupational therapists, music therapists, and sports trainers as participants in this study is well-justified, as these healthcare professionals play complementary roles in the rehabilitation process, each bringing unique expertise that can provide valuable insights into the potential applications of the organ pedal intervention. Physiotherapists specialize in restoring physical function and mobility; occupational therapists focus on enabling individuals to participate in meaningful daily activities; music therapists utilize the therapeutic potential of music to address a wide range of needs; and sports trainers optimize physical performance and prevent sports-related injuries. This multidisciplinary approach ensures that the intervention is assessed from diverse professional perspectives, thereby enhancing the validity and applicability of the findings in both clinical and community settings.

## Ethics approval

Ethical approval was obtained from The Hong Kong Polytechnic University's research ethics committee prior to patient recruitment (ref. no.: HSEARS20230912002).

## Data collection

Focus group interviews were conducted with 4–6 participants per group at a university venue in Hong Kong. Each focus group session lasted approximately 1 hour and 15 minutes and followed a structured flow. The session began with a 5-minute demonstration of the organ pedalboard by a professional concert organist, EF, followed by a 10-minute period where participants had the opportunity to personally try out the instrument. The majority of the session, spanning 50–60 minutes, was then dedicated to a semi-structured discussion guided by open-ended questions. This discussion-focused portion allowed the researchers to thoroughly explore the participants' perceptions, attitudes, and perceived barriers to using the organ pedalboard in rehabilitation settings. The semi-structured approach provided flexibility to probe deeper into relevant topics as they arose.

Semi-structured interviews were conducted by the lead researcher, MK, who has experience in conducting qualitative research in rehabilitation, using open-ended questions developed by the research team. MK facilitated focus group discussions with EF, who clarified organ-specific inquiries. MK asked follow-up questions to prompt responses and seek clarification from participants. All interviews were audio-recorded with written consent. No repeat interviews were carried out. Brief notes were made during interviews.

The interview questions were guided by the Biopsychosocial Model, which provided a conceptual framework for understanding organ pedal training's potential benefits. This model helped structure the inquiry into how physical, psychological, and social factors might influence the effectiveness of the intervention. The questions were designed to comprehensively explore these dimensions, keeping the discussion focused and relevant to the study's objectives (see S2 File). The interview guide was not pilot tested.

## Organ pedalboard description

A standard 32-key radiating concave American Guild of Organists-approved pedalboard was demonstrated. The organ pedalboard used in this study spanned a range from C2 to G4, encompassing 32 musical notes on a standard organ. The lowest note was C2, while the highest reached G4, covering approximately two and a half octaves. This extensive range enabled a variety of foot movements and exercises targeting diverse muscle groups in the lower extremities.

The pedalboard was equipped with a MIDI (Musical Instrument Digital Interface) function, a standard technology that allows electronic musical instruments, computers, and other equipment to communicate and synchronize with each other. In this study, the MIDI function connected the pedalboard to a laptop and an external sound module, enabling it to produce sounds through speakers that mimicked a traditional pipe organ. This feature was crucial in facilitating a realistic musical experience during the rehabilitation exercises. Fig 1 presents the modified pedalboard used in this study.

## Data analysis

An inductive content analysis approach was used to analyse the qualitative data. The inductive approach was employed due to the absence of prior knowledge about the organ pedal training method. This approach could lead to the generation of new concepts and information through data analysis [28, 29].

The analysis process was guided by the Framework Method, a systematic approach that involves identifying similarities and differences in qualitative data and allowing data analyses within and between participants [30, 31]. The seven analytical stages of the Framework Method include (1) transcription, (2) familiarization, (3) coding, (4) development of a working analytical framework, (5) indexing, (6) charting, and (7) interpretation [30].

First, one of the researchers (MK) transcribed all interviews verbatim. The transcripts were not returned to the participants for comments and/or corrections. Then, MK and co-researcher WK (PhD), a male research fellow experienced in qualitative research, independently familiarized themselves with the data through repeated readings, noting initial impressions. Then, they independently applied open codes, line-by-line, before categorizing them using a jointly developed tree diagram. Subsequently, each code and category were indexed as an abbreviation. Nvivo software version 12 was used for stages three to five. Then, the data was reduced and imported into a matrix on an Excel spreadsheet. Each participant was assigned to one row, and each code was placed in a column. Lastly, themes were generated by comparing the similarities and differences between the data on the matrix and interpreted by establishing connections to the data. The participants did not provide feedback on the findings.

Peer checking was employed in this study to enhance the rigor and credibility of the research findings. The researchers MK and WK, both with substantial experience in qualitative research, independently analyzed the data. They engaged in regular discussions throughout the analysis process to compare their interpretations, resolve discrepancies, and refine the coding framework. These peer checking sessions helped ensure that the findings were robust, consistent, and free from individual bias. Additionally, the co-researcher (FA) provided critical feedback on the themes and interpretations, contributing to the overall trustworthiness of the study.

## Reflexivity

The research team, led by MK, who has a background in nursing with a Master's degree in psychology of music, was deeply aware of how personal experiences and professional background could influence the research process. The team consciously reflected on their assumptions about the use of music and organ pedalboards in rehabilitation. Recognizing that their professional backgrounds could predispose them to view these interventions positively, the researchers employed strategies such as peer checking and collaborative data analysis to challenge their interpretations and ensure a more balanced analysis. This reflexive approach contributed to a deeper understanding of how the researchers' perspectives might have shaped the study's design, data collection, and analysis. By acknowledging these influences, the research team

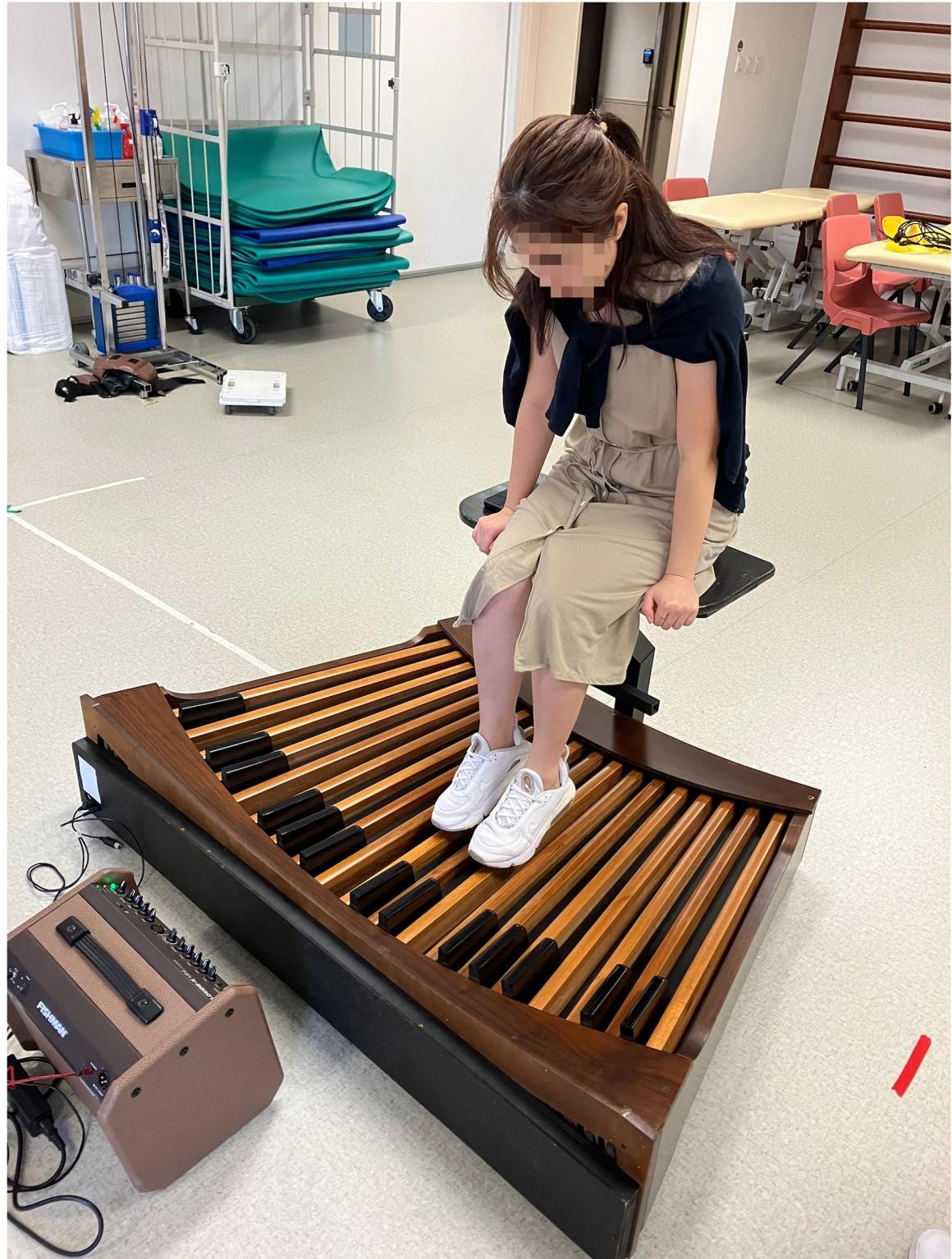

**Fig 1. The modified pedalboard with a MIDI function.**

enhanced the study's transparency, thereby increasing the credibility and reliability of the findings.

Reflexivity was integral to the ethical conduct of this research, as it helped minimize power dynamics and ensure the participants' voices were accurately represented, leading to balanced and objective conclusions. Given that some participants were personally acquainted with the lead researcher (MK), particular measures were employed to minimize potential bias. All participants were explicitly informed of the study's objectives and the distinct role of MK as a facilitator, separate from any personal connection. Participants were encouraged to provide honest and candid feedback, independent of their relationship with MK. Additionally, transparency was reinforced by emphasizing the structured nature of the interview process, which strictly adhered to the interview guide developed by the research team. During the interviews, MK maintained a neutral stance and carefully managed her influence to minimize bias.

## Findings

A total of 17 healthcare providers who agreed to participate in the study attended one of the four focus groups. No participant was dropped out. The focus groups were composed as follows: three groups included one music therapist, one occupational therapist, one physiotherapist, and one sports trainer. One group included two occupational therapists in addition to the other three disciplines, resulting in a total of 17 participants. This composition ensured the representation of a variety of perspectives from different healthcare disciplines were represented.

Nine participants were males. The median age of participants was 32.5 years, with an interquartile range (IQR) of 21 (50–29) years, indicating a relatively broad age distribution. Similarly, the median years of professional practice were 5 years, with an IQR of 6.5 (9.5–3) years. This distribution suggests a diverse range of experience levels among participants, with both early-career and more experienced professionals represented.

Most participants (n = 15, 88.2%) held a Master's degree or higher level of education. The majority of the participants (70%, n = 12) were employed full-time across a variety of healthcare settings, including hospitals (n = 6), elderly care homes (n = 6), private clinics (n = 2), and other facilities (n = 3). The participants primarily served older adult (n = 10), adult (n = 5), and paediatric (n = 2) patient populations. Table 1 presents the demographic characteristics of the focus group participants.

The analysis of the focus group data revealed three higher-order themes that provide a structured framework to better categorize and clarify the sub-themes that emerged. These higher-order themes enhance the readability and understanding of the research findings.

The first higher-order theme is "**The Multifaceted Role of Music in Rehabilitation,**" which encapsulates the overarching influence of music within the rehabilitation context. It includes sub-themes that explore the role of music as a key motivator, encouraging sustained patient engagement and natural movement. Additionally, this theme examines how the organ pedalboard is perceived not only as a therapeutic tool but also as a musical instrument, making the rehabilitation process more engaging and less clinical for patients.

The second higher-order theme is "**Benefits of Organ Pedalboard,**" which brings together various benefits of using the organ pedalboard in rehabilitation. Sub-themes under this category include physical benefits, such as improvements in range of motion, flexibility, balance, and coordination; cognitive benefits, including enhancements in memory, attention, and cognitive processing; as well as the psychosocial benefits, where the pedalboard boosts patient confidence, provides a sense of accomplishment, and facilitates social interaction.

**Table 1. Demographic characteristics of the participants (N = 17).**

| Participant Code | Discipline | Job Status | Age | Gender | Education | Year of Practice | Work setting | Population of Patients | Focus group |
|---|---|---|---|---|---|---|---|---|---|
| MT01 | music therapist | Part-time | 54 | F | Master's | 1 | elderly home | older adults | 1 |
| MT02 | music therapist | Full-time | 34 | F | Master's | 4.5 | elderly home | older adults | 2 |
| MT03 | music therapist | Full-time | 55 | M | Doctoral | 20 | private clinic | older adults | 3 |
| MT04 | music therapist | Full-time | 34 | F | Master's | 10 | elderly home | older adults | 4 |
| OT01 | occupational therapist | Full-time | 30 | F | Master's | 9 | hospital (inpatient) | older adults | 1 |
| OT02 | occupational therapist | Full-time | 34 | M | Master's | 9 | elderly home | older adults | 2 |
| OT03 | occupational therapist | Full-time | 28 | F | Bachelor's | 5 | hospital | children | 3 |
| OT04 | occupational therapist | Part-time | 29 | F | Master's | 6.5 | hospital | adults | 4 |
| OT05 | occupational therapist | Part-time | 29 | F | Master's | 5.5 | hospital | older adults | 4 |
| PT01 | physiotherapist | Full-time | 26 | M | Master's | 3 | hospital | older adults | 1 |
| PT02 | physiotherapist | Part-time | 62 | M | Master's | 39 | elderly home | older adults | 2 |
| PT03 | physiotherapist | Full-time | 31 | M | Master's | 0.5 | home care | older adults | 3 |
| PT04 | physiotherapist | Full-time | 50 | F | Doctoral | 28 | hospital | adults | 4 |
| ST01 | sports trainer | Full-time | 28 | M | Master's | 2 | on-field support (sports) | adults | 1 |
| ST02 | sports trainer | Full-time | 31 | M | Sub-degree | 5 | school | children | 2 |
| ST03 | sports trainer | Part-time | 25 | M | Master's | 3 | elderly home | adults | 3 |
| ST04 | sports trainer | Full-time | 33 | M | Master's | 4 | private clinic | adults | 4 |

The third higher-order theme is "**Modifications of Organ Pedalboard,**" which addresses the logistical and safety aspects of incorporating the organ pedalboard into rehabilitation settings. This theme includes sub-themes that discuss the unique requirements and adaptations necessary for effective use of the pedalboard across different patient populations, the importance of safety features and modifications to prevent injury, and the need for programming and customization to make the pedalboard a more accessible and effective tool for rehabilitation settings.

## Theme 1: The Multifaceted Role of Music in Rehabilitation

**Sub-theme 1.1: Music facilitates rehabilitation.** Participants viewed music as motivating for patients during rehabilitation activities. It allows patients to move naturally by following the inherent musical beats and rhythms. Incorporating music into rehabilitation may encourage patients to engage for sustained periods of time. As one occupational therapist noted:

*"As PT and OT, we focus on physical and cognitive functions; both domains require prolonged training time to achieve training outcomes. It (music) has the potential to encourage the patients to persist in the rehabilitation training."*

*(OT04)*

The healthcare providers discussed neurophysiological mechanisms by which music stimulates the brain and triggers responses facilitating physical and cognitive rehabilitation. A music therapist explained:

*"As a neurologic music therapist, we often treat patients with neurologic injuries, often stroke and Parkinson's patients. Our brain has an uncontrollable physiological reaction to music; important thing is how we can maximise this reaction to help the rehabilitation process, no matter whether physical or speech."*

*(MT04)*

Participants also selected music purposefully for rehabilitation sessions based on target populations and occasions. Familiar music from a patient's generation or related to cultural festivals was believed to evoke memories and associations. As one occupational therapist described:

*"We will play old songs of their generation. Perhaps discuss with them their past memories to stimulate their memory. Or when playing some games related to festivals, for example, Chinese New Year, New Year-related songs will be played to stimulate their memories."*

*(OT2)*

**Sub-theme 1.2: Detaching patient roles through music rehabilitation.** If an organ pedalboard is served as a rehabilitation tool, patients may perceive it as a musical instrument rather than a traditional treatment method. Engaging in training on the organ pedalboard allows patients to view their participation as a music activity, which may help them distance themselves from the awareness of their illness and reduce the perception of returning for rehabilitation treatments.

*"I think music can facilitate the participation of the participants. And they may not think that they are coming for treatment or training. That is, having a feeling of participating in music activities, playing instruments, singing, or different music activities, their participation or feeling is better."*

*(MT03)*

*"They won't feel they only come for treatment; they remember the illness, remember the pain. When they listen to music, they may be more comfortable. Or when they listen to familiar music, for example, when the elderly listen to some old songs, they may have better motivation."*

*(ST03)*

Compared to conventional lower limb rehabilitation interventions, training with the organ pedalboard may offer an alternative and potentially more engaging approach. This can mitigate the boredom that some patients may experience during traditional rehabilitation.

*"I think if the patient is bored when receiving conventional rehab, incorporating musical elements could make the activity more interesting."*

*(ST04)*

## Theme 2: Benefits of Organ Pedalboard

**Sub-theme 2.1: The organ pedalboard as a physical rehabilitation tool.** Healthcare providers who have participated in this study and experienced playing on the organ pedalboard

perceive it as an effective tool for training various aspects of physical rehabilitation, including range of motion, flexibility, balance, and coordination. They identified a wide range of movements that can be performed on the pedalboard involving hip flexion/extension, adduction/abduction, internal/external rotation, knee flexion/extension, dorsiflexion, and plantar flexion.

*"First is mobility range of motion; they will perform a lot of movements like hip flexion, extension, adduction/abduction, external rotation, internal rotation, and knee flexion/extension. And ankle dorsiflexion and plantar flexion as well. When these movements are performed repetitively, it is more than just mobility training but also strength and endurance training and core muscles. I have just tried, and when I hold it (the bench), the legs are only moving. If I am not holding it (the bench) or holding one side only, it utilizes a lot of core (muscles). When it is played repetitively, the waist is quite tiring. So, it is training the core stability, and coordination, and as said, it can be agility training if performing the movement quickly."*

*(PT01)*

*"I think it could improve mobility, flexibility, and balance, but it won't have a significant impact on muscle strength since it would require resistance. I think I can improve my balance."*

*(OT05)*

However, agility, strength, and cardiovascular endurance components can be trained on the pedalboard as well when adapting training elements. For example, agility and cardiovascular endurance can be trained if the training intensity and dosage are increased. Patients' strength and power can be trained by using resistance bands and weights.

*"Based on my experience when playing on the pedalboard, I would say it could have an effect on mobility, flexibility, and strength could also be possible, but not on a cardiovascular level. Perhaps it benefits the cardiovascular system if you train the patients very quickly."*

*(PT04)*

*"I think resistance bands could be added to the exercise so that other movements could also be trained, such as dorsiflexion/plantar flexion and abduction/adduction, but not only 'air resistance' (no resistance)."*

*(ST04)*

*"For endurance, it can be trained as well. For example, if you ask the patient to step from right to left and left to right 10 times, it may be tiring for the elderly. It depends on how the (training) program is set. For power, if you want to train their legs, will you hang a sandbag (on the calf)? These variations, small elements, in answering your questions about how it can influence health."*

*(ST03)*

Healthcare providers also considered alternative movements that patients may naturally adopt to compensate for certain limitations. For instance, patients may use plantar movements instead of engaging their hips fully when stepping on the keys. To target specific movements, such as hip abduction or adduction, it is recommended to introduce obstacles between the keys to encourage the desired motions.

*"Actually, as we said, some movements are targeting the hip; it depends on whether they will actively move up the hip to do so. If I think from the perspective of patients, if I ask you to step keys from the left to the right, I will not follow your instruction of moving up the hip to step the keys one by one. Unless you add some obstacles between keys requiring them to cross over them, they will intentionally move up the hip. If I am patient, I can step on the key next to it without moving the hip up. So, this is the effects whether it can target the hip."*

*(ST03)*

**Sub-theme 2.2: The organ pedalboard as a cognitive rehabilitation tool.** In addition to the physical benefits, healthcare providers also discussed the potential of using organ pedalboards as a cognitive training tool for various patient groups. Learning to step on specific keys requires attention and engages cognitive processes such as sequential and visual memory.

*"I think it is easier to adapt (pedal training) to them (elderly). And as said, training following the orders can help to train memory, sequential memory, etc."*

*(OT02)*

*"I believe the demand for cognitive functioning could be high. They have to read instructions; perhaps it can be facilitated by programming. The good side is they can be trained on sequencing, visual memory, short-term memory, working memory, etc., or attention, etc. for children to stroke patients to elderly; it is helpful."*

*(OT03)*

Pedal training can be particularly beneficial for children with attention deficit hyperactive disorders (ADHD) and mental retardation (MR). When the training is programmed to meet such needs, MR children can benefit from memory training, whereas impulse control can be enhanced in ADHD children.

*"Give you an example: ADHD children are very impulsive. Maybe you have to say that if there is a bubble popping out, they step on the keys when the bubble moves to a certain position. This requires reactive inhibition. They have to wait for a while and step on the pedal until the bubble moves to the designated position. For MR children, mental retard children, they have poor memory; their visual memory is very poor. Say, red, yellow, blue—they have to remember. disappear. They have to step in red, yellow, and blue. It can train working memory a lot and help them a lot. Music or pedal becomes the medium, but not for lower extremities specifically."*

*(OT03)*

**Sub-theme 2.3: The organ pedalboard as a psychosocial tool.** Beyond physical and cognitive benefits, the organ pedalboard can also serve as a valuable psychosocial tool. Patients with upper-limb mobility problems can use the pedalboard to create music using their feet, providing them with a sense of accomplishment and boosting their confidence.

For patients who are unable to play the piano due to hand injuries, the pedalboard offers an alternative means of musical expression. By transferring their skills to their feet, they can regain a sense of confidence and achievement. This unique use of the pedalboard can have a positive impact on their psychological well-being.

*"The pedalboard could also be used not only for clients with lower extremity problems but also for those with upper extremity problems. The patient might be a pianist; he can transfer the skill to playing with feet. I think organ pedalboards are uncommon in the musician's world. So, when a patient with upper extremity problems could complete tasks with lower limbs, they will have a better quality of life and psychological health from a sense of achievement, building up confidence, and identity reassurance."*

*(MT04)*

The pedalboard facilitates social participation and engagement. Patients can engage in rehabilitation activities on the pedalboard with their caregivers or family members. They can work on the movements together and create music collaboratively. This not only strengthens their social relationships but also enhances their overall quality of life.

*"Another method I want to explore is using the pedalboard together in couples; for example, if either one has a stroke or spinal cord injury, they could complete the tasks together, which can improve their relationship, thereby improving the quality of life."*

*(ST04)*

Nevertheless, it is important to note that the effectiveness of the pedalboard as a psychosocial tool may be influenced by individual preferences and sensitivities to music. The sound produced from the pedalboard may be 'noise' for them, leading to frustration.

*"There might be two sides that the pedal exercise might have on one's psychological health or quality of life. As MT (MT04) said, if the patient likes music, the effect should be positive, like a sense of accomplishment; however, if the patient is not sensitive to music, then it could lead them to frustration by not being able to complete the tasks."*

*(ST04)*

### Theme 3: Modifications of Organ Pedalboard

**Sub-theme 3.1: Special circumstances in using the organ pedalboard.** Pedal training can be effectively used during the post-surgery acute phase, even when patients are under bandage or have undergone internal fixation. By having patients seated on a bench, training enables early mobilisation of the lower limb or other body parts that are not involved in the surgery. This encourages active movements, reduces swelling, and promotes recovery.

*"This could also help patients who have just received surgery and are recovering in an acute stage. Usually, the patient has to be immobilized for a period of time after surgery. But in the acute phase, we want to encourage them to engage in active movements to reduce swelling. It can be used in the early stages to help them develop active movement. Often, they are afraid to move body parts after surgery; say, even after ankle surgery, they are afraid to move the hip and knee. Pedal training can be used on body parts that are not involved in injury or surgery. They can still exercise the joints when bandaging or having internal fixation in the acute phase."*

*(OT04)*

Interestingly, a healthcare provider suggested a unique approach by placing the pedalboard vertically or hanging it on the wall. This allows patients who are seated in wheelchairs to enjoy

playing music using their hands. Additionally, the pedalboard can also be played while standing, maximizing upper arm range and movements.

*"They don't have to sit on this bench. And I think, if hung (on a wall) for the upper limbs, the effectiveness may be even greater than stepping with feet. The upper limb can be trained when seated or standing. And the range of movement for the hands is better than that of the hip. Different things can be played. It is different from your initial thought. Because you originally were targeting the feet."*

*(PT02)*

**Sub-theme 3.2: Ensuring safety when using the organ pedalboard as a rehabilitation tool.** Healthcare providers emphasized the importance of implementing safety measures when using the pedalboard for rehabilitation purposes. The current organ bench design is not optimal for rehabilitation use, so it is recommended to adjust the chair height and position to fit the patient's ergonomics. Additionally, a rotational chair with handrails and a footstep can be considered to facilitate the transfer of weaker patients to the instrument for rehabilitation exercises.

*"This can be solved if the bench first has a handrail. Secondly, it can be rotated. Rotate it to the side so the patient can sit on it and then rotate it back to the position to play the pedalboard that serves the function (purpose)."*

*(PT02)*

*"Back support, straps, handle—if the patient needs to receive these exercises, it means their balance is very weak. And when without back support, moving up the legs to reach out keys requires high trunk stability. . . And the strap prevents them from slipping down. Make sure these must be done. Also, regarding safety, get on and off the seat. The patient may not be able to get on (the seat) themselves. Maybe a rotational seat? A step or an inclined platform to reach the seat? These have to be sorted out."*

*(PT04)*

Healthcare providers expressed concerns about the level of supervision and manpower required when prescribing pedal exercises to their patients. However, they stated that if armrests, back support, and a seatbelt or pelvic belt are added to prevent patients from slipping from the chair, they would be confident in allowing patients to practice the movements on the pedalboard independently.

*"If patients' trunk support is weak, can a back support be added? If they cannot sit stable, can they add a seat belt? Add a handrail at the side. I suppose patients who use it may be those who are more capable. But if what I suggested can be added, those who are less capable can move their legs and enjoy the fun of music as well."*

*(OT01)*

*"As said, regarding the balancing problem, if one is seated without a backrest, from the therapist's perspective, we are not confident enough to let them. . . It turns out it's a safety issue. If there is backrest and they have suboptimal balance, we are confident to let them sit on it to press and step (the pedalboard)."*

*(ST03)*

*"Use some methods, e.g., a monitor, and follow the monitor to do movements—this kind of method. The level of supervision and assistance can be lowered. That is, fasten a seat belt and let them sit on it stably; they can do it themselves."*

*(PT02)*

**Sub-theme 3.3: Programming the organ pedalboard to serve as a rehabilitation tool.**
One of the featured elements of the organ pedalboard is music. While organists read sheet music, patients and healthcare providers who are not musicians need a simple cueing system to guide the rehabilitation movements in creating music. By implementing relevant programming, the organ pedalboard can become a feasible tool for rehabilitation training.

Some healthcare providers suggested programming the lighting on the pedal keys. Patients can step on the designated keys when the light is flashing. Or else, using numbers and colours as cues instead of reading musical notations.

*"For those who have little musical knowledge or are not playing this instrument, some things can be added. Perhaps add light bulbs to step on it when it is lit. This makes the pedal a tool that can be used wider and broader."*

*(ST01)*

*"You have to think about whether it can be used by all rehabilitation workers. If you want to involve music, there should be some music resources. How you design it so it can be used by rehabilitation workers who do not know music, such as numbering, colouring, but not a score. You should have a wide variety of repertoire."*

*(MT04).*

Nevertheless, a music therapist suggested retaining the option of music scores for those experienced musicians.

*"I think both are needed. Some participants do not know music, they want to follow the song, colour, this kind. For those who have music experience, they can read the score to play songs themselves."*

*(MT03)*

Healthcare providers suggested signal cues on a screen monitor. Patients could follow the instructions displayed on the screen to perform the corresponding movements.

*"The problem is how to cue the patients to follow your movements. Have you ever considered using a screen monitor to display, use the light signal, or step on the one that is lighting? These adaptations have to be made. If they sit on it but don't know how to play, the training becomes non-constructive. The best is that they can follow your instructions. And think of how you can make them follow."*

*(PT02)*

Furthermore, the selection of songs and sounds should be programmed to cater to the diverse needs and abilities of the patients. Patients may listen to a soundtrack and perform

simple movements on the pedalboard. Both the sound and the patient's playing can be programmed so that they can listen through earphones, avoiding disturbances for other patients in the rehabilitation setting.

> *"Or the patients can participate by playing a few notes in sync with the beat of a song. The music library can be modified; programming for drum sound. Not sure if the patients know this is a pipe organ. They may want to listen to organ sounds. This will be left to patients' choice."*
>
> *(OT01)*

> *"Because of the (limited) space and also the noise, not everyone likes it. As you said, it can be used with earphones, but it is associated with safety issues. If the surrounding noises are covered with safety, someone calls you, suddenly there is fire safety, etc., it is one of the concerns."*
>
> *(ST03)*

## Discussion

We addressed the research objectives outlined in the introduction and successfully identified healthcare providers' perceptions of using organ pedalboards as a rehabilitation tool for lower extremity conditions. The findings revealed that healthcare providers view music as a motivator that enhances patient engagement and supports natural movement, particularly in lower limb rehabilitation. Additionally, the study highlighted key barriers and facilitators influencing the adoption of organ pedalboards in clinical practice, such as safety concerns and the need for customized training programs. Importantly, the healthcare providers provided specific recommendations on how organ pedal training could be integrated into existing rehabilitation protocols, emphasizing its potential to improve physical, cognitive, and psychosocial outcomes. These insights contribute significantly to the development of targeted rehabilitation strategies and underscore the need for further research to optimize the use of organ pedalboards in clinical settings.

The findings of this study provide valuable insights into healthcare providers' perceptions and attitudes towards the use of organ pedalboards as a rehabilitation tool for lower extremity conditions. The healthcare providers' views generally align with previous research on the role of music in rehabilitation. Music has been widely recognized as a motivating factor for patients during rehabilitation activities [32, 33]. It allows patients to move naturally by following the inherent rhythm and beats of the music, encouraging sustained engagement [34].

Researchers have extensively studied and documented the therapeutic role of music in rehabilitation. It has been shown to motivate patients to participate in rehabilitation activities and facilitate natural movements [35]. Music has also been found to stimulate the brain and trigger physiological responses, aiding in physical and cognitive rehabilitation [36]. Furthermore, the use of familiar music can trigger patients' memories and associations, potentially enhancing the overall rehabilitation experience [37]. These findings emphasize the potential of music intervention in diverting attention from discomfort and reducing perceived exertion levels, highlighting its prospective utility in rehabilitation settings [38].

The use of the organ pedalboard was perceived as an innovative approach to rehabilitation, offering an alternative to traditional methods. This aligns with the growing body of literature advocating for the integration of creative and engaging methods in rehabilitation to enhance patient adherence and outcomes [39]. The participants thought that patients may positively

perceive training on the organ pedalboard as a music activity, potentially leading to increased participation and sustained engagement in rehabilitation [40]. The participants suggested that the organ pedalboard could be used to train a variety of physical aspects, including range of motion, flexibility, balance, and coordination. This is consistent with previous research that has demonstrated the effectiveness of music-based interventions in improving these physical attributes [41, 42]. Moreover, the pedalboard can facilitate social participation and engagement, as patients can engage in rehabilitation activities with their caregivers or family members. This collaborative activity can strengthen social relationships and enhance the quality of life [25]. However, individual preferences and sensitivities to music should be considered, as the sound produced from the pedalboard may be perceived as 'noise' by some patients, leading to frustration [43].

Interestingly, the participants also identified the potential of the organ pedalboard as a cognitive rehabilitation tool. This is a novel finding that warrants further investigation. The use of the organ pedalboard could potentially enhance cognitive processes such as attention and memory, which are crucial for successful rehabilitation [44–46]. The potential benefits are not only limited to adults but also to other patient groups, including children with special educational needs such as ADHD and MR [47]. This finding suggests a promising avenue for future research.

The organ pedalboard has been identified by the participants as a potential psychosocial tool for rehabilitation in patients with upper or lower limb problems. The ability to create music using the pedalboard can provide a sense of accomplishment and confidence, which can be beneficial for patients' mental health and overall well-being [25]. Accumulating evidence shows that music interventions have a significant effect on patients by reducing their level of pain [48], anxiety [49], and stress [50].

However, the study also identified potential challenges in implementing organ pedal training, such as the need for specific adaptations to target certain movements and the potential for patients to adopt compensatory movements. These findings highlight the importance of careful planning and individualized programming in the implementation of organ pedal training. Furthermore, it is essential to address the safety considerations associated with using the organ pedalboard as a rehabilitation tool. The current design of the organ bench is not optimal for rehabilitation use. Adjustments to the chair height and position, as well as the addition of armrests, back support, and a seatbelt or pelvic belt, can prevent patients from slipping from the chair [51]. Additionally, the level of supervision and manpower required for patients practicing movements on the pedalboard should be carefully considered, along with the implementation of safety measures such as back support, straps, and handlebars to prevent slips and falls [52]. Furthermore, the programming of the organ pedalboard for rehabilitation purposes is crucial, with the need for cueing methods, signal displays, and song and sound selection to meet the diverse needs and abilities of patients [53]. Programming the organ pedalboard can make it a feasible tool for rehabilitation training. Simple cueing systems, such as lighting on the pedal keys or signal cues on a screen monitor, can guide the rehabilitation movements in creating music. Visual feedback has been largely used in rehabilitation interventions, as it has been shown to be effective [54, 55].

This study highlights the importance of aligning healthcare professionals' perspectives with the proposed theoretical framework for the use of organ pedalboards in rehabilitation. The findings reveal that professionals' insights generally support its key components, particularly in relation to the biopsychosocial model. This model emphasizes the interplay of biological, psychological, and social factors in health, which are also reflected in the perceived benefits of organ pedal training. The professionals agreed that the organ pedalboard could effectively address the biological aspects of rehabilitation by improving range of motion, flexibility, and

coordination. These practical insights align with the framework's biological dimension, affirming its relevance in real-world applications. Additionally, the psychological and social benefits of using the organ pedalboard, such as enhanced patient motivation, cognitive engagement, and social participation, also resonate with the framework's focus on these areas. However, some discrepancies between theoretical expectations and practical challenges were identified. For example, while the framework posits that music-based interventions could universally enhance emotional well-being, the professionals noted that patient preferences for music vary significantly, which could impact the tool's effectiveness. This feedback suggests a need for further refinement of the framework to account for individual differences and responses to music. Incorporating these practical experiences into the framework would enhance its applicability and credibility, ensuring it addresses real-world challenges while maintaining theoretical rigor. Aligning the framework with the insights of regular users increases the likelihood of its successful adoption and effective use in rehabilitation settings, ultimately leading to better patient outcomes.

The cost of implementing organ pedalboards in rehabilitation is indeed a critical factor that must be acknowledged. While our study focused on the potential benefits of this tool, the oversight of not emphasizing the high cost, as highlighted in the interview guide, should be addressed. The participants' primary focus was on the therapeutic advantages rather than the financial barriers. It was because most of the participants worked for an organization. They used tools provided by the organization to manage their patients. Understanding the therapeutic effects of pedalboards for patients enabled participants to recommend this tool to the decision makers of their workplace. However, the modification and programming costs associated with adapting the pedalboard for rehabilitation could pose significant challenges, particularly in resource-limited settings. To mitigate these concerns, it is important to clarify that the pedalboard is not intended for individual patient ownership but rather should be purchased and maintained by institutions or clinical settings, such as hospitals and research institutes, where patients would undergo training. Future research could investigate the feasibility of this institutional ownership approach. This strategy is anticipated to enhance accessibility and alleviate individual financial burdens, allowing the primary focus to remain on the clinical benefits of this rehabilitation tool.

The cultural context must be carefully considered when evaluating the adoption of the organ pedalboard as a rehabilitation tool. However, we did not discuss any cultural factors influencing its adoption during this exploratory study due to time constraints. In the Hong Kong settings where the study was conducted, the organ (pedalboard) is not a popular musical instrument. This lack of familiarity may create cultural barriers, as patients may prefer to work with instruments they are more accustomed to, such as the piano or traditional Chinese instruments. These cultural preferences could impact the effectiveness of the intervention, as patient engagement is closely linked to their comfort and familiarity with the therapeutic tools used. However, it is important not to make assumptions that patients would be unwilling to try the pedalboard. In fact, the novelty of this "new instrument" may actually create a sense of intrigue and interest, potentially facilitating greater patient engagement. The current unpopularity of the pedalboard might be due to factors such as its availability, personal interest, or lack of exposure and training, rather than inherent cultural aversions. We recommend that future research assess the potential effectiveness of integrating the organ pedalboard into rehabilitation settings under institutional ownership, along with the implementation of 'train-the-trainer' programs for healthcare professionals. By offering the pedalboard as an option, patients can explore this unique tool as part of their rehabilitation, potentially enhancing their overall experience and outcomes. Importantly, future research should also explore patients' perceptions, attitudes, and experiences with the organ pedalboard directly. Understanding the

target population's perspectives, preferences, and cultural considerations will be crucial in developing effective strategies for the successful integration of this rehabilitation tool.

The findings offer valuable insights, but they should be interpreted within Hong Kong's sociocultural context. We believe that the findings of the study can also be applicable to the regions with sociocultural contexts similar to Hong Kong. Generalizability to communities with different cultural backgrounds or healthcare systems may be limited, as Hong Kong is a developed region where Western therapies are widely accepted. However, future research should explore the applicability in diverse cultural settings to better understand the broader potential and limitations of this approach. Such cross-cultural investigations will provide crucial insights into how the organ pedalboard intervention may need to be tailored or adapted to ensure its effectiveness and acceptability in varied settings.

Our study has the strength of having four different types of healthcare professionals involved. They provided detailed expert comments and opinions about the study topic comprehensively. A notable limitation is that the interview guide lacked pilot testing before implementation. This could have a missed chance to refine questions or structure, potentially affecting data quality and depth. Future studies should pilot the interview guide to identify and address issues prior to full data collection. Another limitation of the current study is the absence of comprehensive field notes taken by a dedicated note-taker during the interviews. While the researchers made brief notes throughout the discussions, these did not capture the level of detail and contextual observations that formal field notes would have provided. The lack of a dedicated note-taker may have resulted in the loss of valuable insights into the non-verbal cues, interactions, and nuances of the interview process. Meanwhile, member checking was not conducted due to time constraints. This may limit the ability to confirm whether our interpretations and findings accurately reflect participants' viewpoints.

Other limitations are that we did not apply modifications to the organ accessories, such as chairs and lights for cueing, and pilot them on lower-limb patients.

We also did not gather voices and opinions from the perspectives of patients and organizations. While patients are the end-users, we are uncertain how they perceive this novel tool adopted in their rehabilitation. We are also concerned about the acceptability of policymakers regarding the use of musical instruments for rehabilitation purposes. In the future study, we aim to modify the organ accessories and software program and test its feasibility through a pilot study on patients with particular lower limb problems. We will also conduct qualitative studies, collecting the perceptions of patients, healthcare providers, and policymakers on the implementation and actual use of the tool (instrument) in rehabilitation settings.

## Conclusion

Our study is the first to explore the feasibility of using a pedalboard as a rehabilitation tool. The introduction of the organ pedalboard as a rehabilitation tool presents a promising avenue for lower extremity rehabilitation. Its potential to improve the physical, cognitive, and psychosocial aspects of rehabilitation underscores its significance in the field of healthcare. However, further research and development are necessary to address safety concerns and program the pedalboard for rehabilitation. This will assist in the development of a lower extremity training protocol that can accommodate all the physiological hip, knee, ankle, and foot movements in the future.

## Supporting information

**S1 File. COREQ (COnsolidated criteria for REporting Qualitative research) checklist.** (DOCX)

**S2 File. Interview guide.**
(DOCX)

## Author Contributions

**Conceptualization:** Mandy M. P. Kan, Eric C. Fan, Fadi M. Al Zoubi.

**Data curation:** Mandy M. P. Kan, Eric C. Fan.

**Formal analysis:** Mandy M. P. Kan, Fadi M. Al Zoubi.

**Investigation:** Mandy M. P. Kan, Eric C. Fan, Fadi M. Al Zoubi.

**Methodology:** Mandy M. P. Kan, Eric C. Fan, Fadi M. Al Zoubi.

**Project administration:** Mandy M. P. Kan.

**Resources:** Mandy M. P. Kan, Eric C. Fan, Fadi M. Al Zoubi.

**Supervision:** Fadi M. Al Zoubi.

**Validation:** Fadi M. Al Zoubi.

**Writing – original draft:** Mandy M. P. Kan, Fadi M. Al Zoubi.

**Writing – review & editing:** Mandy M. P. Kan, Wai Hang Kwok, Fadi M. Al Zoubi.

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
