## [Decision Letter · Decision Letter 0]

8 Aug 2024

PONE-D-24-14802Organ pedalboard as a rehabilitation tool: a qualitative exploratory study of healthcare providers’ perceptions and recommendationsPLOS ONE

Dear Dr. Al Zoubi,

Thank you for submitting your manuscript to PLOS ONE. After careful consideration, we feel that it has merit but does not fully meet PLOS ONE’s publication criteria as it currently stands. Therefore, we invite you to submit a revised version of the manuscript that addresses the points raised during the review process.

 The reviewers noted a number of points which altogether could improve your manuscript. In particular, I wish to highlight their comments in relation to a theoretical/conceptual framework which I hope you would consider an important aspect of this revision. Please submit your revised manuscript by Sep 22 2024 11:59PM. If you will need more time than this to complete your revisions, please reply to this message or contact the journal office at plosone@plos.org. Please include the following items when submitting your revised manuscript:A rebuttal letter that responds to each point raised by the academic editor and reviewer(s). You should upload this letter as a separate file labeled 'Response to Reviewers'.A marked-up copy of your manuscript that highlights changes made to the original version. You should upload this as a separate file labeled 'Revised Manuscript with Track Changes'.An unmarked version of your revised paper without tracked changes. You should upload this as a separate file labeled 'Manuscript'.

We look forward to receiving your revised manuscript.

Kind regards,

Catherine M. Capio

Academic Editor

PLOS ONE

Journal Requirements:

**Additional Editor Comments:**

We had now secured two reports on your manuscript and both reviewers raised important points that need to be addressed. Crucially, both reviewers noted that a theoretical/conceptual framework is an important aspect that the authors should address which I would like to emphasise.

Reviewers' comments:

Reviewer's Responses to Questions

**Comments to the Author**

1. Is the manuscript technically sound, and do the data support the conclusions?

Reviewer #1: Yes

Reviewer #2: Yes

2. Has the statistical analysis been performed appropriately and rigorously? 

Reviewer #1: N/A

Reviewer #2: Yes

3. Have the authors made all data underlying the findings in their manuscript fully available?

Reviewer #1: Yes

Reviewer #2: Yes

4. Is the manuscript presented in an intelligible fashion and written in standard English?

Reviewer #1: Yes

Reviewer #2: Yes

5. Review Comments to the Author

Reviewer #1: I would like to thank the authors for reporting such an interesting and novel study. I read through the manuscript with great interest, and I think it has the potential to extend our knowledge about music-based rehabilitation with organ pedalboard. However, due to some issues listed below, there is still room for improvement in the manuscript.

1. Background, theoretical framework (if any), and rationale for the study can be strengthened.

The authors have listed out some previous findings about the treatment effects of music-based intervention for rehabilitation. However, it remains unclear about why the authors would like to target lower extremity conditions, which seemed to be a relatively broad category. It would also be important to put more emphasis on the review of previous findings on existing music-based intervention for lower extremity conditions. This can help to identify the research gap.

Similarly, it would be essential to state why organ pedalboard was chosen as a potential rehabilitation tool. What makes it unique or more suitable than other musical instruments?

In addition, the rationale for selecting physiotherapists, occupational therapists, music therapists, and sports trainers as participants should be added. It would also be helpful to provide background information about the rehabilitation work of these four groups of health professionals in clinical and community settings so that readers may differentiate them from each other.

2. Issue about research objectives.

The authors may note that the objectives stated in Abstract (Lines 25-28) are not fully consistent with those in Introduction (Lines 90-95). Also, the section of Discussion should start with a paragraph to summarize major findings in response to each research objective.

3. Method

The rationale for adopting focus group interviews should be provided. The author may also explain why each focus group included different healthcare providers, instead of having focus groups with providers from the same discipline. What group dynamics were expected and observed?

During each focus group interview, it seemed that a considerable proportion of time was spent on demonstration and trials of organ pedalboard. How long was the discussion among participants?

As some participants knew the lead researcher personally, the authors may disclose more details about personal reflexivity regarding this point.

Since “lower extremity conditions” is a broad concept, it could be difficult to reach data saturation. It would be helpful if the authors can report how data saturation was achieved.

The study appeared to be a semi-structured focus group interview, rather than a structured interview.

4. Findings

Following COREQ, please name the section as “Findings” instead of “Results”.

The major themes emerged from the focus group interviews are nicely presented. It would be better to identify sub-themes so that readers can capture the findings more easily.

Please consider reporting participants job status (full-time or part-time) and the population of their patients in Table 1.

5. Discussion

According to the interview guide, the high cost of organ pedalboard was listed as a potential barrier. It seemed counter-intuitive that this was not included in Findings. Moreover, the modification and programming of organ pedalboard to address the barriers would result in even higher cost. The authors may discuss this point.

Also, since organ pedalboard is not a commonly used musical instrument, could there be cultural barriers in adopting it for rehabilitation? Would patients prefer musical instruments that they are familiar with?

The authors may also try to interpret the findings in the sociocultural context of Hong Kong and see if they are culturally specific or generalizable to other communities.

Reviewer #2: August 7, 2024

Dear Authors,

Thank you for the opportunity of reading your manuscript titled “Organ pedalboard as a rehabilitation tool: a qualitative exploratory study of healthcare.” The primary aim of the study is to explore the feasibility of using organ pedal training as a rehabilitation tool for patients with lower extremity problems. The researchers sought to understand healthcare providers' perceptions regarding the use of organ pedalboards in rehabilitation, including the potential benefits, attitudes, and perceived barriers associated with this innovative approach. The study's findings reveal that healthcare providers view organ pedalboards as a promising rehabilitation tool, particularly for lower extremity rehabilitation. They perceive music as a significant motivator that facilitates natural movement through rhythm and beats, enhancing patient engagement and psychosocial well-being. The pedalboard training is reported to improve range of motion, balance, and coordination, while also offering cognitive benefits. However, safety concerns regarding its implementation were acknowledged, highlighting the need for careful planning. Overall, the insights from healthcare providers can inform the development of a structured training protocol that accommodates various physiological movements, thereby enhancing rehabilitation practices.

The study is notable for its innovative approach in exploring the use of organ pedalboards for rehabilitation. It comprehensively examines physical, cognitive, and psychosocial benefits, involving diverse healthcare providers to gather a wide range of insights. The qualitative methodology allows for in-depth understanding, while practical recommendations address safety and usability concerns. By identifying barriers and facilitators, the study provides valuable guidance for clinical implementation and sets a foundation for future research, potentially benefiting a broad range of patients.

The paper would benefit from the inclusion of a conceptual framework, a section on reflexivity, and pieces of evidence to enhance the credibility of the results. This can be achieved through member checks or peer debriefing. Please review the comments provided below.

Comment on the Introduction

1. Could you provide information about the study design and numerical results? It will give the readers idea on the quality of evidence being presented. At the point in the text that reads “another study reported enhanced muscle endurance in middle-aged adults performing lower limb exercises while listening to music [15]. These findings underscore the potential of music intervention to divert attention from discomfort and reduce perceived exertion levels [16-18], highlighting its prospective utility in rehabilitation settings.”

2. Could you please provide evidence, whether direct or indirect, as reported in the literature, on the effectiveness of organ pedaling in treating lower extremity conditions?

3. Please provide a conceptual framework or theoretical model (e.g., Normalization Process Theory, Theoretical Framework of User Experience, Biopsychosocial model) to propose the beneficial effects of organ pedaling on lower extremity conditions. Including a conceptual framework in qualitative studies is essential for several reasons. It guides the research process, offering a clear structure for conducting the study and illustrating the relationships between key concepts and variables, thus ensuring focus and coherence. Furthermore, it clarifies the underlying assumptions, theories, and beliefs that shape the research, providing important context and perspective. The conceptual framework influences the choice of research methods and techniques, aligning them with the research objectives and questions. It also strengthens the validity and credibility of the study by grounding it in existing knowledge and theory, making it easier to justify and explain the findings. Additionally, it aids in interpreting the findings by providing a lens through which the data can be understood, linking the results to broader theories and concepts, and making them more meaningful and relevant. Moreover, it communicates the scope and boundaries of the study to the audience, preventing misinterpretations. Finally, it helps identify gaps in existing knowledge and literature that the study aims to address, situating the research within the broader academic discourse and demonstrating its contribution. Overall, a conceptual framework is a foundational element in qualitative research that ensures the study is systematic, coherent, and grounded in existing knowledge and theory.

Comments on the Method

1. What are the eligibility criteria regarding experience in utilizing an organ pedalboard? Is it deemed essential?

2. How was the relationship between these participants and MK, whom they personally know, addressed? At the point in the text that reads “Few initial participants knew MK personally, a female research associate (registered nurse; MA in psychology in music) prior to the research.

3. Have the participants signed an informed consent form? At the point in the text that reads “Participants and sampling.”

4. A conceptual framework will be useful in understanding the reasons behind the questions during the interview. At the point in the text that reads “Structured interviews were conducted by the lead researcher, MK, who has experience in conducting qualitative research in rehabilitation, using open-ended questions developed by the research team.”

5. This should be included as a study limitation. At the point in the text that reads “The interview guide was not pilot tested.”

6. Was there any field note-taker during the interview?

7. What does C2-G4 mean?

8. What does MIDI mean?

9. Was member checking conducted? Member checking, also referred to as participant validation or respondent validation, is a technique employed in qualitative research to bolster the credibility and accuracy of the findings. During this procedure, researchers share their interpretations, analyses, or summaries of the data with the participants who contributed the data. The objective is to confirm that the researchers' depictions accurately reflect the participants' experiences, viewpoints, and interpretations. If member checking was performed, please describe the process.

10. Was peer checking performed? Peer checking, also known as peer debriefing or peer review, is a process used in qualitative research to enhance the rigor and credibility of the study. This technique involves researchers sharing their data, analysis, or interpretations with colleagues or peers who have expertise in the field or in qualitative research methods. The purpose is to gain additional perspectives and ensure that the research process and findings are accurate, reliable, and free from bias. If it was performed, how?

Comments on the Results

1. What is the homogeneity of the age and year of practice distribution? Should the median (IQR) be reported instead of the mean (SD)?

2. Please revise this that it clearly adds up to 17 participants. At the point in the text that reads “Each group consisted of one MT, one OT, one PT, and 1 one ST, except for one group, which included two OTs.”

3. Did the investigators obtain information regarding the participants' areas of expertise?

4. What type of rehabilitation tool is this? Does it address emotional needs? At the point in the text that reads “Utilizing the music/organ pedalboard as a rehabilitation tool.”

Comments on the Discussion

1. It would have been interesting to determine if professionals' perspectives regarding the use of organ pedalboards align with the proposed framework, which is not presented in the Introduction. Understanding professionals' views on the use of organ pedalboards in relation to the proposed framework is essential for several reasons. Firstly, if professionals agree with the framework, it affirms the framework's relevance and effectiveness in practical settings. Their insights can reveal practical challenges, benefits, and nuances that may not be immediately evident from a purely theoretical standpoint. By incorporating these practical experiences, the framework can be refined to better address real-world applications. Moreover, professionals' feedback can lead to meaningful improvements and innovations, ensuring the framework remains relevant and effective. This alignment not only enhances the credibility of the framework but also increases the likelihood of its adoption and effective use within the field. In conclusion, incorporating the perspectives of regular users of organ pedalboards guarantees that theoretical frameworks are both academically rigorous and practically valuable.

Suggestion on Reflexivity

1. It would be beneficial to include a section on reflexivity. Reflexivity in qualitative research involves the process of reflecting on and critically examining how researchers’ own backgrounds, perspectives, and interactions with the research process influence their study. This entails recognizing and disclosing potential biases, assumptions, and personal experiences that could shape the research design, data collection, analysis, and interpretation. Reflexivity is crucial because it provides transparency about the researcher's role and potential biases, thereby enhancing the credibility and trustworthiness of the research. By acknowledging their own biases and preconceptions, researchers can better account for how these factors might impact their findings and interpretations, leading to more balanced and objective analysis. Additionally, reflexivity promotes ethical considerations by addressing power dynamics and the researcher-participant relationship, encouraging a more respectful and mindful approach to research. Reflexivity is typically addressed in the Methodology section of a manuscript, where researchers discuss their background and how it might influence the research process. However, it can also be mentioned in the Introduction to provide context or revisited in the Discussion section to reflect on how the researcher’s perspectives impacted the findings. Overall, reflexivity is a critical aspect of qualitative research as it ensures that the study is conducted with awareness and integrity, contributing to more credible and insightful results.

I pray the comments help in improving the manuscript. I look forward to reading the updated manuscript.

Stay Blessed,

The Reviewer

6. PLOS authors have the option to publish the peer review history of their article (what does this mean?). If published, this will include your full peer review and any attached files.

Reviewer #1: No

Reviewer #2: **Yes: **Valentin C. Dones III

---

## [Author Response · Author response to Decision Letter 0]

17 Sep 2024

Thank you for reviewing our manuscript and providing constructive feedback to improve it. We have addressed the comments point-by-point and submitted a revised version with and without tracked changes. The key change we made was to incorporate a Biopsychosocial Model to guide our study.

---

## [Decision Letter · Decision Letter 1]

25 Oct 2024

PONE-D-24-14802R1Organ pedalboard as a rehabilitation tool: a qualitative exploratory study of healthcare providers’ perceptions and recommendationsPLOS ONE

Dear Dr. Al Zoubi,

Thank you for submitting your manuscript to PLOS ONE. After careful consideration, we feel that it has merit but does not fully meet PLOS ONE’s publication criteria as it currently stands. Therefore, we invite you to submit a revised version of the manuscript that addresses the points raised during the review process.

 The revised manuscript thoroughly addressed the points raised by the reviewers. While one reviewer had recommended the paper to be accepted, one other reviewer have a few minor suggestions left. I tend to agree that those suggestions could help improve the clarity of your paper, which you could probably address fairly quickly.

We look forward to receiving your revised manuscript.

Kind regards,

Catherine M. Capio

Academic Editor

PLOS ONE

Journal Requirements:

Additional Editor Comments:

We have now received the reports of the reviewers on your revised manuscript. All the points that were previously raised by the reviewers had been thoroughly addressed. There are a few minor suggestions from one reviewer, and I tend to agree from an editorial point of view that these suggestions will improve the clarity of your message.

Reviewers' comments:

Reviewer's Responses to Questions

**Comments to the Author**

1. If the authors have adequately addressed your comments raised in a previous round of review and you feel that this manuscript is now acceptable for publication, you may indicate that here to bypass the “Comments to the Author” section, enter your conflict of interest statement in the “Confidential to Editor” section, and submit your "Accept" recommendation.

Reviewer #1: All comments have been addressed

Reviewer #2: All comments have been addressed

2. Is the manuscript technically sound, and do the data support the conclusions?

Reviewer #1: Yes

Reviewer #2: Yes

3. Has the statistical analysis been performed appropriately and rigorously? 

Reviewer #1: N/A

Reviewer #2: Yes

4. Have the authors made all data underlying the findings in their manuscript fully available?

Reviewer #1: No

Reviewer #2: Yes

5. Is the manuscript presented in an intelligible fashion and written in standard English?

Reviewer #1: Yes

Reviewer #2: Yes

6. Review Comments to the Author

Reviewer #1: I would like to thank the authors for their efforts in the revision. The manuscript has been improved significantly in terms of conceptual base, methodological credibility, clarity of findings, and critical discussion. Here are a few further points for authors’ consideration:

1. The identification of higher-order themes helped to present the findings clearly. I may suggest that the higher-order themes can be more concise. For example, the second higher-order theme can be “Benefits of Organ Pedalboard”, and the third one can be “Modifications of Organ Pedalboard”. Also, the wordings in the two sub-themes under the first higher-order theme appear to be similar. It is not easy to differentiate these two without reading the text and quotes.

2. The discussion on the issue of cost (Lines 773-784) is critical and valid. However, it seems more important to discuss why this theme was not extensively discussed during the focus group interviews and how future studies may address this. The authors mentioned that it was probably due to “participants’ primary focus…on the therapeutic advantages”. It is important to interpret why they had such primary focus. Possible influential factors may include the project team’s underlying assumptions of therapeutic effects of organ pedalboard, limited time/session for focus group interviews, or the role of the participants (as health professionals rather than managers of health services). It is a good suggestion of applying organ pedalboard through institutional ownership, but may consider soften the tone, making this a possible direction for future study.

3. Like the issue of cost, the authors may also interpret why cultural barriers (Lines 786-805) was not covered in the findings of the study. Also, may consider soften the tone about the point of “train-the-trainer”, making it a possible future direction of research.

4. In the discussion of generalizability (Lines 807-815), the authors mentioned that “We believe that there are minimal barriers to pedalboard rehabilitation adoption in similar developed regions globally”, and this seems stretching too far. The findings of the study can be applicable to the regions with sociocultural context similar to Hong Kong. But it does not mean that organ pedalboard can be applied successfully, with minimal barriers.

5. Here may be some typos:

Lines 131-139: “The repetitive goal-oriented nature to improve functional outcomes through tasks that mimic daily activities” Do you mean “The repetitive goal-oriented nature can improve functional outcomes through tasks that mimic daily activities” ?

Lines 807-815: “The findings offer valuable insights, but they should be interpreted within Hong Kong's sociocultural.” Do you mean “…but they should be interpreted within Hong Kong's sociocultural context” ?

Reviewer #2: Thank you for thoroughly addressing the provided comments and suggestions. I recommend that your revised manuscript be accepted for publication in PLOS ONE. Congratulations!

7. PLOS authors have the option to publish the peer review history of their article (what does this mean?). If published, this will include your full peer review and any attached files.

Reviewer #1: No

Reviewer #2: **Yes: **Valentin C. Dones III

---

## [Author Response · Author response to Decision Letter 1]

28 Oct 2024

Dear reviewers,

We would like to thank you again for taking the time to review our manuscript and provide constructive comments to facilitate the revision of our manuscript. We addressed the additional comments from one of the reviewers and made minor revisions on the manuscript. 

All the best,

---

## [Editor Report · Decision Letter 2]

6 Nov 2024

Organ pedalboard as a rehabilitation tool: a qualitative exploratory study of healthcare providers’ perceptions and recommendations

PONE-D-24-14802R2

Dear Dr. Al Zoubi,

We’re pleased to inform you that your manuscript has been judged scientifically suitable for publication and will be formally accepted for publication once it meets all outstanding technical requirements.

Kind regards,

Catherine M. Capio

Academic Editor

PLOS ONE
---

## [Editor Report · Acceptance letter]

11 Nov 2024

PONE-D-24-14802R2 

PLOS ONE

Dear Dr. Al Zoubi, 

I'm pleased to inform you that your manuscript has been deemed suitable for publication in PLOS ONE. Congratulations! Your manuscript is now being handed over to our production team.

Kind regards, 

on behalf of

Dr. Catherine M. Capio 

Academic Editor

PLOS ONE